# Relationship between Harsh Parenting and Aggressive Behaviors in Male Juvenile Delinquents: Potential Mediating Roles of Peer Victimization and Hostile Attribution Bias

**DOI:** 10.3390/bs13070610

**Published:** 2023-07-22

**Authors:** Shuang Lin, Ying Wang, Gonglu Cheng, Xuejun Bai

**Affiliations:** 1Key Research Base of Humanities and Social Sciences of the Ministry of Education, Academy of Psychology and Behavior, Tianjin Normal University, Tianjin 300074, China; lins000@126.com (S.L.); chgonglu@126.com (G.C.); 2Faculty of Psychology, Tianjin Normal University, Tianjin 300387, China; 3Faculty of Education, Guangxi Normal University, Guilin 541004, China; wangying202304@126.com

**Keywords:** harsh parenting, aggressive behavior, peer victimization, hostile attribution bias, male juvenile delinquent

## Abstract

Harsh parenting can be regarded as a harsh behaviors, feelings, and attitudes toward children in the process of parenting. According to the theory of intergenerational transmission of violence, harsh parenting is an important factor affecting children’s aggressive behavior, but the theory does not clarify the specific action path between harsh parenting and aggressive behavior. In order to reveal the relationship between harsh parenting and the aggressive behavior of juvenile delinquents, 604 male juvenile delinquents (N = 604; M_age_ = 16.57 years; SD = 0.612 years) were investigated using the Harsh Parenting Questionnaire, the Buss–Perry Aggression Questionnaire, the Multidimensional Peer Victimization Scale, and the Word Sentence Association Paradigm for Hostility in Chinese. Analysis using structural equation modeling procedures showed that (a) all variables were positively associated with each other; (b) the partial indirect effect of harsh parenting on aggressive behaviors was determined through the mediators of peer victimization and hostile attribution bias independently; and (c) the partial indirect effect was determined through the mediators of peer victimization and hostile attribution bias sequentially. The results suggest that harsh parenting can explain the highly aggressive behavior of male juvenile delinquents. Moreover, harsh parenting may also predict the risk of peer victimization and hostile attribution bias, thereby predicting the development of highly aggressive behaviors.

## 1. Introduction

Adolescence is an important stage of transition and development in life. Aggressive behavior presents a gradually increasing trend with the entry of individuals into adolescence and reaches a peak in adolescence [1,2]. Aggressive behavior is any behavior directed toward another individual that is carried out with the proximate (immediate) intent to cause harm [3]. Aggressive behavior not only violates social norms but also has negative effects on physical and mental health, academic progress, emotional regulation, behavior shaping, personality development, and social adaptation after adulthood [4]. More dangerously, aggressive behavior may increase the risk of crime in male adolescents [5]. A survey’s results showed that the amounts and severity of aggressive behavior in juvenile delinquents are higher than that in common adolescents of the same age [6]. Moreover, the data in China’s White Paper on Juvenile Prosecutorial Work (2021) showed that in 2021, the number of juvenile suspects reviewed and prosecuted by the procuratorial organ included 19,061 for larceny; 9049 for affray; 7591 for rape; 7186 for robbery; and 6902 for provocation as the top five, accounting for 25.8%, 12.2%, 10.3%, 9.7%, and 9.3% of the total, respectively, and four of the five types involve violent or aggressive behavior [7].

According to the theory of intergenerational transmission of violence, harsh parenting may explain why children show severe aggressive behavior and even violent behavior [8]. In particular, harsh parenting may transmit an aggressive pattern of interpersonal interaction to children, who easily internalize this pattern into their own behavior and apply it to a broader process of interpersonal interaction after long-term reinforcement of this pattern [9]. Harsh parenting refers to the harsh treatment of children in the process of parenting, including through behavior, emotion, and attitude [10]. Harsh parenting includes physical aggression in the form of spanking, slapping, pinching/twisting, and hitting with objects; verbal aggression in the form of abuse, sarcasm, and scolding; mental aggression in the form of ignorance, neglect, and exclusion; and over-control in the form of supervision and obedience. The significant positive correlation between harsh parenting and aggressive behavior in elementary, middle, and college students has been widely discussed [11,12,13]. However, there are few studies on the relationship between harsh parenting and juvenile delinquents’ aggressive behavior. Considering the importance of clarifying the causes of the violent criminal behavior of juvenile delinquents, the current study aimed to explore the relationship between harsh parenting and the aggressive behavior of juvenile delinquents. Therefore, we propose the first hypothesis that harsh parenting can directly and positively predict aggressive behavior (H1).

However, the theory of intergenerational transmission of violence has not clarified the specific action path of harsh parenting on children’s aggressive behavior. According to the social information processing model [14], harsh-parenting parents are unable to show reasonable emotional and behavioral control strategies to children, which may lead to emotional disorders and impulsivity in children, forming aggressive hostile attribution bias, and responding with irrational aggressive behaviors. Parents’ physical and verbal aggression might cause children to be overly vigilant to potentially threatening social cues, develop hostile attribution bias, and thus experience difficulty controlling angry responses and exhibiting aggressive behavior [15]. Hostile attribution bias is a tendency to attribute hostile intentions to peers in ambiguous circumstances [16]. Although existing studies have not directly explored the mediating role of hostile attribution bias between harsh parenting and aggressive behavior, the relationship between harsh parenting and hostile attribution bias, and the relationship between hostile attribution bias and aggressive behavior has been fully verified in previous studies. For example, Milner et al. (2017) conducted six studies demonstrating that reducing harsh parenting can reduce children’s hostile attribution bias [17]. Perhamus and Ostrov (2021) showed that children’s hostility attribution bias could positively predict their subsequent aggressive behavior in a longitudinal study [18]. Accordingly, we propose the second hypothesis that hostile attribution bias plays a mediating role in the relationship between harsh parenting and aggressive behavior (H2).

According to the developmental cascade model of adolescent aggression [19], the evolution of individual aggression or violent behavior is influenced by early family factors (e.g., harsh parenting) and school factors (e.g., peer victimization); moreover, family factors have a progressive influence on individuals through school factors. In particular, after experiencing harsh parenting, children transfer the negative emotions learned from parent–child interactions to peer situations, which leads to poor peer relationships. In sum, the cumulative negative experiences at the family and school levels further magnify the individual’s cognitive bias and, eventually, lead to the outbreak of severe aggressive behavior. Peer victimization is defined as physical or psychological injury from peers in the forms of physical aggression, verbal aggression, and relational aggression [20]. While previous studies have not specially examed the connection between harsh parenting and aggressive behavior through the pathways of peer victimization and hostility attribution bias, it is possible that physiological or neurological disorders resulting from harsh parenting may provide an explanation for this relationship. For example, Lewis et al. (2021) found that children who experience long-term exposure to harsh parenting tend to remain highly alert to anger signals during adolescences. This heightened alertness triggers the individual’s sympathetic-adrenal-spinal system and HPA axis system, leading to the frequent release of catecholamine and cortisol [21]. Following psychological and physiological changes, adolescents who have experienced harsh parenting display heightened sensitivity to stressful situations, such as peer victimization. This, in turn, exacerbates the hostile attribution bias and leads to the expression of anger through aggressive behavior. Therefore, we propose the third and fourth hypotheses: (H3) peer victimization plays a mediating role in the relationship between harsh parenting and aggressive behavior; and (H4) peer victimization and hostility attribution bias play a chain mediating role between harsh parenting and aggressive behavior.

## 2. Materials and Methods

### 2.1. Participants

A total of 630 male juvenile delinquents were randomly recruited from juvenile prisons in a Chinese province. The participants’ age ranged from 15 to 17, with a mean age of 16.57 years (SD = 0.61). The mean age at which they entered prison was 16.31 (SD = 0.78). Among the participants, 221 (36.6%) had at least one parent who had migrated for work, resulting in them being stay behind. After eliminating the answers of participants who did not answer carefully or missed more than half of the questions, the final valid sample size was 604, corresponding to a retention rate of 95.87%. The main types of crimes committed by these juvenile delinquents include robbery (51%), rape (25%), intentional injury (13%), theft (2%), intentional homicide (2%), etc. On average, they committed 1.03 crimes (SD = 0.21) and had an initial term of sentence with a mean of 3.63 years (SD = 2.61).

### 2.2. Measures

Harsh parenting. Harsh parenting was assessed by the Chinese version [22] of the Harsh Parenting Questionnaire [23]. This questionnaire consisted of 4 items (e.g., “Dad hit me or kicked me.”) rated on a five-point scale ranging from 1 (strongly disagree) to 5 (strongly agree). A high mean score indicates serious harsh parenting for all items. The structure validity of the Chinese version was found to be good in the current study, as confirmed by the results of confirmatory factor analysis (χ^2^/df = 43.41, CFI = 0.96, TLI = 0.92, SRMR = 0.03, RMSEA = 0.08, 95%CI RMSEA = [0.06, 0.11]). The Cronbach′s alpha coefficient for the total questionnaire was 0.77, while for the father’s and mother’s versions, it was 0.69 and 0.67, respectively.

Aggressive behavior. Aggressive behavior was assessed using the Chinese version [24] of the Buss-Perry Aggression Questionnaire [25]. The sub-scale of physically aggressive behavior (e.g., “Once in a while, I cannot control the urge to strike another person.”) and the sub-scale of verbally aggressive behavior (e.g., “I tell my friends openly when I disagree with them.”) were used to measure aggressive behavior. A high mean score on all items indicates serious aggressive behavior. Confirmatory factor analysis was conducted to assess the structure validity of the Chinese version, yielding good results (χ^2^/df = 66.191, CFI = 0.93, TLI = 0.91, SRMR = 0.03, RMSEA = 0.04, 95%CI RMSEA = [0.03, 0.05]). The Cronbach′s alpha coefficient for the sub-scale of physically aggressive behavior was 0.61, while for the sub-scale of verbally aggressive behavior, it was 0.64. The total questionnaire had a Cronbach’s alpha coefficient of 0.68.

Peer victimization. As sub-scales of the Chinese version of the Multidimensional Peer-Victimization Scale [26], physical victimization (e.g., “Beat me up.”) and the sub-scale of relational victimization (e.g., “Tried to make my friends turn against me.”) were used to assess peer victimization. Serious victimization was indicated by a high mean score for all items. In the current study, confirmatory factor analysis was conducted to assess the structure validity of the Chinese version. The results showed good validity (χ^2^/df = 149.80, CFI = 0.93, SRMR = 0.04, RMSEA = 0.06, 95%CI RMSEA = [0.05, 0.08]). The Cronbach’s alpha coefficient for the sub-scale of physical victimization was 0.63, while for the sub-scale of verbal victimization, it was 0.80. The total questionnaire had a Cronbach’s alpha coefficient of 0.81.

Hostile attribution bias. Hostile attribution bias was assessed using the Chinese version [27] of the Word Sentence Association Paradigm for Hostility [16]. The sub-scale of hostile attribution bias, which contained 16 distinct ambiguous sentences (e.g., “Someone is in your way”), followed by either a hostility-related word (e.g., “inconsiderate”) or a benign word (e.g., “unaware”). Each sentence was randomly presented twice. A high mean score on the sub-scale indicated a high level of hostile attribution bias for all items. The data were subjected to confirmatory factor analysis, which demonstrated good structural validity for the Chinese version (χ2/df = 141.78, CFI = 0.93, TLI = 0.92, SRMR = 0.04, RMSEA = 0.03, 95%CI RMSEA = [0.02, 0.04]). The Cronbach′s alpha coefficient for the sub-scale was 0.72.

### 2.3. Procedure and Statistical Analysis

Questionnaires were distributed to each block in cooperation with the prison guards, with the assistance of two doctoral psychology students. These participants were instructed to read the instructions carefully before completing the scale. During the testing process, if the subjects had any doubts about the questionnaire questions, they were allowed to ask the interviewer questions at any time. The test lasted for approximately 30 min, and the questionnaires were collected by the researcher upon completion. The study was approved by the Ethics Committee of the Faculty of Psychology, Tianjin Normal University. All participants signed a written consent form prior to the study. The data were subsequently analyzed using SPSS 18.0 and Mplus 7.0 through descriptive analysis, correlation analysis, and multiple mediation analysis.

## 3. Results

### 3.1. Common Method Bias

Harman’s single-factor test was utilized to test the common method bias when the data were collected. The results revealed that 13 factors exhibited an eigenvalue exceeding 1. However, the initial factor solely accounted for 12.41% of the total variance, which falls beneath the critical standard of 40%. Hence, no significant common method bias was observed.

### 3.2. Descriptive Statistics and Correlations

In Table 1, we display the means, standard deviations, and correlations of the research variables. For the demographics, the results of correlational studies revealed several significant correlations between demographic variables, such as age and initial term of sentence, age and age to enter prison, degree of education and age to enter prison, degree of education and mothering harsh parenting, type of crime and count of crime, type of crime and age to enter prison, type of crime and mothering harsh parenting, count of crime and initial term of sentence, count of crime and fathering harsh parenting. For the study variables, there were significant positive correlations among one another.

### 3.3. Mediation Model Analysis

The hypothesized mediation model was examined using SEM. The final model, which is presented in Figure 1, exhibited a good fit with the data, CFI = 0.98, TLI = 0.94, RMSEA = 0.06, and SRMR = 0.06. The results of bootstrap test showed that the direct effect of harsh parenting on aggressive behavior was positive and significant (β = 0.28, *p* < 0.001, 95% CI [0.15, 0.42]), and the total indirect effect of harsh parenting on aggressive behavior via the two mediators were positive and significant (β = 0.1, *p* < 0.001, 95% CI [0.06, 0.16]). The mediating effects of peer victimization and hostile attribution bias are presented in Table 2 and Figure 1.

## 4. Discussion

In this study, we sampled male juvenile offenders as research subjects to investigate the relationship between harsh parenting and aggressive behavior. Previous research has mainly focused on general samples of children, adolescents, and adults rather than specifically examining individuals with severe aggression. Our findings indicate that harsh parenting directly predicts aggressive behavior in juvenile delinquents. Additionally, we identified the partial mediating effect of peer victimization and hostile attribution bias between harsh parenting and aggressive behavior. Importantly, we also observed a chain mediating effect, whereby peer victimization and hostile attribution bias mediate the relationship between harsh parenting and aggressive behavior.

The finding that harsh parenting might positively predict aggressive behavior is consistent with previous studies and confirms H1. It suggests that the more serious the harsh parenting experienced by juvenile delinquents, the easier it is for them to observe, learn, and imitate their parents’ aggressive behavior, and that they may then apply those scripts to interpersonal communication. For instance, Liu et al. (2022) recruited 235 Chinese adolescents as participants to investigate the relationship between harsh parenting and aggressive behavior [13]. They found that harsh parenting significantly predicted aggressive behavior among children. Similarly, Cortes Hidalgo et al. (2022) used rest-state fMRI to scan 2410 children at age 10 who experienced maternal harsh parenting, and they observed smaller total gray, cerebral white matter, and amygdala volumes, which are associated with aggressive behavior [28]. Additionally, preventing harsh parenting can decrease aggressive behavior in children. Milner et al. (2017) showed that using evaluative conditioning (EC) improved parents’ attitudes towards upbringing, reduced the demand for educating children, and avoided harsh parenting [17]. These changes can foster positive attachment between parents and children, leading to a warmer perception of society and reducing the inclination towards violence. However, there is currently limited research on the relationship between harsh parenting and aggressive behavior among juvenile delinquents. Future studies should focus on investigating this link among juvenile delinquents, adult delinquents, and individuals prone to high aggression. Longitudinal research is particularly needed to explore the family factors contributing to aggressive behavior in these populations.

Our results supported the dynamic cascade model [19], which suggests that the evolution of individual aggressive behavior or violence is influenced by early factors of family and school. According to this model, factors of family have a progressive impact on individuals through factors of school. In line with this, our second finding revealed a partial mediating effect of peer victimization between harsh parenting and aggressive behavior. Specifically, harsh parenting was found to positively predict peer victimization, which in turn predicted aggressive behavior. This finding supports the H3 hypothesis and is consistent with existing literature. For instance, Perry et al. (2021) conducted a longitudinal study examining the relationships between family violence, peer victimization, and aggressive behavior across different stages of childhood and early adolescence [29]. They found that family violence significantly predicted both peer victimization and aggressive behavior. The inability of children with harsh parenting to establish healthy attachments with their parents may hinder their ability to develop healthy attachments with their peers [30]. Consequently, these children may become more vulnerable to exclusion or even serious violations by other children. Unfortunately, negative experiences from both parents and peers can trigger thoughts of retaliation, leading to aggressive behavior towards these assailants.

Several studies have demonstrated that harsh parenting is a predictor of emotional regulation disorders and selective attention to hostile information, leading to internal problems (such as social anxiety) and external problems (such as aggressive behavior) [31] Especially, parents’ harsh parenting can be viewed as an unsatisfied signal towards children, which induce them to become more sensitive to exclusion, and prone to express hostility in ways of aggressive behaviors or violence. Additionally, Zhao et al. (2021) recruited 76 male juvenile delinquents as participants and confirmed the relation between hostile attribution bias and aggressive behavior [6]. Following up on these findings, our study confirmed the partial mediating effect of hostile attribution bias between harsh parenting and aggressive behavior (H2), which supports the social information processing model [14]. This finding adds to the existing literature, as previous studies have not specifically explored this mediating effect. Given these findings, it is important to recognize the effects of reducing hostile attribution bias in preventing aggressive behavior.

The results of the current study also support the idea of a developmental cascade model of adolescent aggression [19]. The model suggests that negative daily events, such as harsh parenting and peer victimization, activate negative self-schema, leading individuals to feel hostility around the world and initiate negative perspectives about the future. This increases their hostile attribution bias towards the external environment. Additionally, harsh parenting and peer victimization may activate threatening schema and aggravate thoughts of hostility, ultimately enhancing aggression in interpersonal communication. Our study supported this idea and confirmed the chain mediating effects of peer victimization and hostile attribution bias between harsh parenting and aggressive behavior (H4), indicating that harsh parenting positively predicts hostile attribution bias, peer victimization, and, subsequently, aggressive behavior. Individuals who experience harsh parenting tend to be more sensitive to stressful environments, overreacting to environmental stimuli. This can result in peer rejection, victimization, and the formation of a hostile attribution bias towards external environmental information. Ultimately, these individuals may exhibit serious aggressive behavior or even violent criminal behavior [32]. A genetic study conducted by Brody et al. (2014) further suggests that harsh parenting can impair the short allele carried by 5-HTTLPR, leading individuals to overreact to the external environment. In particular, threatening stimuli become a priority and cannot be overridden, increasing the risk of aggressive behavior [33]. It should be noted that previous empirical studies have not explored the chain mediating role of peer victimization and hostile attribution bias between harsh parenting and aggressive behavior.

The current study has a few limitations. First, the study is based on self-reported data, which is susceptible to social desirability biases, particularly given the negative nature of the topics being investigated, such as harsh parenting, peer victimization, hostile attribution bias, and aggressive behavior. Second, the cross-sectional design of the study prevents the examination of causal relationships among the variables in the model. However, it is worth noting that no longitudinal study on the relationship between harsh parenting and aggressive behavior has been conducted to date. Third, the current study only explores the dimensions of physical and verbal aggression from the aggressive behavior form, omitting the investigation of the motivation for aggressive behavior (i.e., proactive aggression and reactive aggression [34]). Future research should consider reporting variables from various perspectives, such as including reports from parents regarding harsh parenting and reports from peers regarding peer victimization. Additionally, future studies should employ longitudinal designs to investigate the long-term effects of harsh parenting on aggressive behavior and the mediating role of peer victimization and hostile attribution bias. It may also be beneficial to include other types of aggressive behaviors to examine the differential influence of different forms of harsh parenting on various aggressive behaviors.

## 5. Conclusions

In the present study, we investigated the relationship between harsh parenting and aggressive behavior in male juvenile delinquents. Our findings indicated that harsh parenting increased the likelihood of aggressive behavior in male adolescents. Specifically, the negative effects of harsh parenting on aggressive behavior could be explained by peer victimization and the development of a hostile attribution bias.

## Figures and Tables

**Figure 1 behavsci-13-00610-f001:**
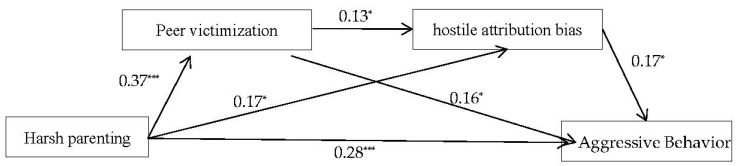
Final model illustrates the chain mediation of peer victimization and hostile attribution bias on the association between harsh parenting and aggressive behavior. * *p* < 0.05; *** *p* < 0.001.

**Table 1 behavsci-13-00610-t001:** Descriptive statistics and correlations of measures (N = 604).

	M ± SD	1	2	3	4	5	6	7	8	9	10	11	12	13	14
1.Age	16.57 ± 0.61	1													
2.DE	0.96 ± 0.42	0.1 *	1												
3.TOC	0.05 ± 0.23	0.01 *	0.01	1											
4.COC	1.03 ± 0.21	0.08	0.05	0.25 ***	1										
5.ITS	3.63 ± 2.61	0.14 ***	0.02	−0.10 *	0.12 ***	1									
6.AEP	16.30 ± 0.78	0.55 **	0.16 **	0.14 ***	0.03	−0.06	1								
7.SD	0.37 ± 0.48	−0.06	−0.02	0.05	0.02	−0.01	−0.01	1							
8.FHP	7.68 ± 2.618	−0.004	0.05	0.14	0.16 ***	−0.02	−0.01	0.04	1						
9.MHP	6.92 ± 2.518	0.00	−0.08 *	0.09 *	0.00	−0.03	−0.02	0.02	0.56 ***	1					
10.HAB	46.43 ± 9.848	−0.06	0.07	−0.01	0.01	−0.03	−0.07	0.03	0.14 ***	0.10 *	1				
11.PAB	16.18 ± 4.227	0.03	−0.02	0.07	0.00	−0.07	0.05	−0.01	0.25 ***	0.19 ***	0.26 ***	1			
12.VAB	13.15 ± 3.091	−0.02	0.04	0.05	−0.05	−0.06	0.01	−0.04	0.18 ***	0.20 ***	0.19 ***	0.46 ***	1		
13.PV	5.96 ± 1.831	−0.02	0.06	0.00	−0.02	0.02	−0.02	0.00	0.22 ***	0.21 ***	0.10 *	0.20 ***	0.23 ***	1	
14.RV	15.50 ± 4.617	−0.002	0.01	−0.05	0.05	0.03	−0.05	0.01	0.16 ***	0.19 ***	0.16 ***	0.17 ***	0.26 ***	0.43 ***	1

Note: * *p* < 0.05; ** *p* < 0.01; *** *p* < 0.001. DE, degree of education; TOC, type of crime; COC, count of crime; ITS, initial term of sentence; AEP, age to enter prison; SD, stayed behind; FHP, fathering harsh parenting; MHP, mothering harsh parenting; HAB, hostile attribution bias; PAB, physical aggressive behavior; VAB, verbal aggressive behavior; PV, peer victimization; RV, relational victimization. Type of crime was coded as 0—violent crimes (e.g., intentional injury, intentional murder, and rape); 1—economic crimes (e.g., stealing, organized prostitution, and drug trafficking); 2—others. Degree of education was coded as 0—primary school; 1—junior high school; 2—senior high school.

**Table 2 behavsci-13-00610-t002:** Testing the mediation effect of HP on AB.

Effect	β	*p*	95% CI	Ratio of Total Effects
Direct Effect				
HP→AB	0.28 ***	<0.001	[0.15, 0.42]	74.32%
Indirect Effect				
HP→PV→AB	0.06 *	<0.01	[0.03, 0.11]	15.95%
HP→HAB→AB	0.03 *	<0.05	[0.01, 0.07]	7.57%
HP→PV→HAB→AB	0.01 *	<0.05	[0.002, 0.02]	2.16%
Total Mediation Effect	0.10 ***	<0.001	[0.06, 0.16]	25.68%

Note: HP, harsh parenting; PV, peer victimization; HAB, hostile attribution bias; AB, aggressive behavior; TIE, total indirect effect. * *p* < 0.05; *** *p* < 0.001.

## Data Availability

The data from this study will not be disclosed in order to protect the privacy of the juvenile delinquents involved. However, if you are interested in accessing the data, you can contact the author.

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
