# Peer review of "Relationship between Harsh Parenting and Aggressive Behaviors in Male Juvenile Delinquents: Potential Mediating Roles of Peer Victimization and Hostile Attribution Bias"

_behavsci, 2023, doi:10.3390/bs13070610_

Round 1

Reviewer 1 Report

Relationship between Harsh Parenting and Aggressive Behaviors among Male Juvenile Delinquent: Potential Mediating roles of Peer Victimization and Hostile Attribution Bias

This paper deals with an interesting topic to be published by Behavioral Sciences. However, there are currently some aspects that need to be reviewed and corrected.

Introduction

- This section must be expanded with ideas that appear in the discussion.

References

- Some references are not included in the corresponding section. Eg "White Paper on Juvenile Prosecutorial Work". Authors must also explain where they get it from, country, ...

- References that appear in the text do not follow the rules of the Journal. Others are indicated by numbering and author name.

- References must be presented in order of appearance and indicating the numbering that appears in the text.

- The name of the Journal must appear abbreviated.

Materials and Methods

- The data provided by the participants is insufficient. It is convenient to expand some sociodemographic data, such as family characteristics, parental education, etc. Above all, because authors are analyzing the type of parenting and the informants are only adolescents. Therefore, authors are analyzing the perception of adolescents about the upbringing received. The authors must justify.

- Authors report that the questionnaires were delivered and collected later. How did you check that the instructions were understood correctly? What happened to the missing data?

- It is not clear that mediation increases the relationship between harsh parenting and aggressive behavior. In fact, in Table 2 the betas are smaller.

Discussion

- Many ideas that appear in this section are not argued in the introductory section. The ideas and references that appear in the discussion must be dealt with previously.

- Some ideas are ambitious. Eg: “The results suggested 281 that the early experience of harsh parenting may enhance the hostile attribution bias and 282 ultimately predict aggressive behavior by increasing the risk of peer victimization”. These ideas should be substantiated by a longitudinal study.

- How did you arrive at this observation? “The finding that harsh parenting might induce aggressive behavior indicated that more serious harsh parenting the juvenile delinquents suffered” How was controlled severity and harsh parenting?

Author Response

Dear reviewer,

We sincerely appreciate your valuable comments and suggestions on our manuscript “Relationship between Harsh Parenting and Aggressive Behaviors among Male Juvenile Delinquent: Potential Mediating roles of Peer Victimization and Hostile Attribution ”(behavsci-2418867). Based on your comments, we have made revision to our manuscript as described below.

Point 1: Introduction: This section must be expanded with ideas that appear in the discussion. 

Response 1: Thanks for your suggestion, we have expanded and reorganized the content of the introduction, as shown in the blue font in the introduction.

Point 2: References: Some references are not included in the corresponding section. Eg "White Paper on Juvenile Prosecutorial Work". Authors must also explain where they get it from, country, ... References that appear in the text do not follow the rules of the Journal. Others are indicated by numbering and author name. References must be presented in order of appearance and indicating the numbering that appears in the text.

- The name of the Journal must appear abbreviated. 

Response 2: Thank you for your comments. We have modified the format of all references according to the requirements of the journal, as shown in the blue part of the paper.

Point 3: Materials and Methods: The data provided by the participants is insufficient. It is convenient to expand some sociodemographic data, such as family characteristics, parental education, etc. Above all, because authors are analyzing the type of parenting and the informants are only adolescents. Therefore, authors are analyzing the perception of adolescents about the upbringing received. The authors must justify.

Response 3: Thank you for your suggestion on increasing the subject information. We have added relevant information about the subject's education level, nationality, staying behind, age of entering prison, number of crimes, and initial term of sentence, as shown in the blue font of the participants part.

Point 4: Materials and Methods: Authors report that the questionnaires were delivered and collected later. How did you check that the instructions were understood correctly? What happened to the missing data?

Response 4: Thank you for your comments on the issue of ensuring that the guidance is properly understood, in the process of testing, if the subjects had doubts about the questionnaire questions, they could ask the interviewer questions at any time. Regarding the missing data, because some participants didn't answer carefully or missed more than half of the questions, their answers were eliminated, and the final valid sample size was 604 (95.87% retention rate). The above statement has been modified in the text, as shown in the blue part of the text.

Point 5: Materials and Methods: It is not clear that mediation increases the relationship between harsh parenting and aggressive behavior. In fact, in Table 2 the betas are smaller.

Response 5: Thank you for your question. We had repeatedly checked the results of the study, and it is true that the effect size of the intermediary effect is small. To ensure a credible result, we further use the PROCESS in SPSS 22.0 to test our indirect effects against signal method bias. By using a bootstraped chain mediation model(using 5000 bias-corrected bootstrap samples; PROCESS, Model 6), we also find significantly indirect effect of Peer victimization (Effect size = 0.02, 95%CI=[0.01, 0.04]) or hostile attribution bias (Effect size = 0.15, 95%CI=[0.09, 0.22]) respectively, as well as a chain indirect effect of Peer victimization and hostile attribution bias (Effect size = 0.05, 95%CI=[0.03, 0.08]) between harsh parenting and aggressive behaviors. Those results support our main mediation analysis in 3.3. Mediation model analysis. We will further verify this problem in future tracking studies. 

Point 6: Discussion: Many ideas that appear in this section are not argued in the introductory section. The ideas and references that appear in the discussion must be dealt with previously.

Response 6: Thanks for your suggestion, we have expanded and reorganized the content of the introduction, as shown in the blue font in the introduction.

Point 7: Discussion: Some ideas are ambitious. Eg: “The results suggested 281 that the early experience of harsh parenting may enhance the hostile attribution bias and 282 ultimately predict aggressive behavior by increasing the risk of peer victimization”. These ideas should be substantiated by a longitudinal study.

Response 7: Thanks for your suggestions, some of the research ideas are really ambitious, so we have revised the expression of the research ideas according to your suggestions, as shown in the blue text in the discussion.

Point 8: Discussion: How did you arrive at this observation? “The finding that harsh parenting might induce aggressive behavior indicated that more serious harsh parenting the juvenile delinquents suffered” How was controlled severity and harsh parenting?

Response 8: Thanks for your question, the statement that harsh parenting might induce aggressive behavior indicated that more serious harsh parenting the juvenile delinquents suffered is not accurate. Therefore, we have revised the statement that harsh parenting positively predicts the aggressive behavior of juvenile delinquents.

We sincerely appreciate your valuable comments and suggestions. Should our revisions be inadequate or incorrect, we kindly ask for the opportunity to further revise the manuscript please. Your feedback is invaluable, and we are grateful for your assistance and support throughout this process. Thank you once again for your guidance and support.

Reviewer 2 Report

The present article has very positive parts, such as investigating whether peer victimization and hostility attribution bias played a chain mediating role between harsh parenting and aggressive behavior.

1. In this manuscript it is not well structured, however, In relation to the method section it should include the following section: Procedure (add how the questionnaires were administered and how it guaranteed the confidential treatment of the data, as well as if the research has the favorable report of any Committee).

2. Regarding the results, it is suggested that you describe in detail in the text the correlations that appear in Table 1.

Congratulations on the work

Author Response

Dear reviewer,

We sincerely appreciate your valuable comments and suggestions on our manuscript “Relationship between Harsh Parenting and Aggressive Behaviors among Male Juvenile Delinquent: Potential Mediating roles of Peer Victimization and Hostile Attribution ”(behavsci-2418867). Based on your comments, we have made revision to our manuscript as described below.

Point 1: In this manuscript it is not well structured, however, In relation to the method section it should include the following section: Procedure (add how the questionnaires were administered and how it guaranteed the confidential treatment of the data, as well as if the research has the favorable report of any Committee).

Response 1: Thanks for your suggestion, we have revised the structure and method of the paper according to your comments, as shown in the blue part of 2. Materials and Methods.

Point 2: In this manuscript it is not well structured, however, In relation to the method section it should include the following section: Procedure (add how the questionnaires were administered and how it guaranteed the confidential treatment of the data, as well as if the research has the favorable report of any Committee).

Response 2:Thanks for your suggestion to describe the correlations in Table 1 in detail, we have described the correlation results in Table 1 in detail, as shown in the blue part of 3.2 . Descriptive statistics and correlations.

We sincerely appreciate your valuable comments and suggestions. Should our revisions be inadequate or incorrect, we kindly ask for the opportunity to further revise the manuscript please. Your feedback is invaluable, and we are grateful for your assistance and support throughout this process. Thank you once again for your guidance and support.
